# The BCAT1 CXXC Motif Provides Protection against ROS in Acute Myeloid Leukaemia Cells

**DOI:** 10.3390/antiox11040683

**Published:** 2022-03-31

**Authors:** James Hillier, Gemma J. Allcott, Laura A. Guest, Wayne Heaselgrave, Alex Tonks, Myra E. Conway, Amy L. Cherry, Steven J. Coles

**Affiliations:** 1School of Science & the Environment, University of Worcester, Worcester WR2 6AJ, UK; j.hillier1@aston.ac.uk (J.H.); allg1_19@uni.worc.ac.uk (G.J.A.); laura.guest-3@postgrad.manchester.ac.uk (L.A.G.); a.cherry@worc.ac.uk (A.L.C.); 2College of Life & Health Sciences, School of Biosciences, Aston University, Birmingham B4 7ET, UK; 3School of Biomedical Science & Physiology, University of Wolverhampton, Wolverhampton WV1 1LY, UK; w.heaselgrave@wlv.ac.uk; 4Department of Haematology, Division of Cancer & Genetics, School of Medicine, Cardiff University, Cardiff CF14 4XN, UK; tonksa@cardiff.ac.uk; 5College of Science & Engineering, University of Derby, Derby DE22 1GB, UK; m.conway2@derby.ac.uk

**Keywords:** BCAT1, CXXC-motif, AML, myeloid, leukaemia, ROS, cysteine, antioxidant

## Abstract

The cytosolic branched-chain aminotransferase (BCAT1) has received attention for its role in myeloid leukaemia development, where studies indicate metabolic adaptations due to BCAT1 up-regulation. BCAT1, like the mitochondria isoform (BCAT2), shares a conserved CXXC motif ~10 Å from the active site. This CXXC motif has been shown to act as a ‘redox-switch’ in the enzymatic regulation of the BCAT proteins, however the response to reactive oxygen species (ROS) differs between BCAT isoforms. Studies indicate that the BCAT1 CXXC motif is several orders of magnitude less sensitive to the effects of ROS compared with BCAT2. Moreover, estimation of the reduction mid-point potential of BCAT1, indicates that BCAT1 is more reductive in nature and may possess antioxidant properties. Therefore, the aim of this study was to further characterise the BCAT1 CXXC motif and evaluate its role in acute myeloid leukaemia. Our biochemical analyses show that purified wild-type (WT) BCAT1 protein could metabolise H_2_O_2_ in vitro, whereas CXXC motif mutant or WT BCAT2 could not, demonstrating for the first time a novel antioxidant role for the BCAT1 CXXC motif. Transformed U937 AML cells over-expressing WT BCAT1, showed lower levels of intracellular ROS compared with cells over-expressing the CXXC motif mutant (CXXS) or Vector Controls, indicating that the BCAT1 CXXC motif may buffer intracellular ROS, impacting on cell proliferation. U937 AML cells over-expressing WT BCAT1 displayed less cellular differentiation, as observed by a reduction of the myeloid markers; CD11b, CD14, CD68, and CD36. This finding suggests a role for the BCAT1 CXXC motif in cell development, which is an important pathological feature of myeloid leukaemia, a disease characterised by a block in myeloid differentiation. Furthermore, WT BCAT1 cells were more resistant to apoptosis compared with CXXS BCAT1 cells, an important observation given the role of ROS in apoptotic signalling and myeloid leukaemia development. Since CD36 has been shown to be Nrf2 regulated, we investigated the expression of the Nrf2 regulated gene, *TrxRD1*. Our data show that the expression of *TrxRD1* was downregulated in transformed U937 AML cells overexpressing WT BCAT1, which taken with the reduction in CD36 implicates less Nrf2 activation. Therefore, this finding may implicate the BCAT1 CXXC motif in wider cellular redox-mediated processes. Altogether, this study provides the first evidence to suggest that the BCAT1 CXXC motif may contribute to the buffering of ROS levels inside AML cells, which may impact ROS-mediated processes in the development of myeloid leukaemia.

## 1. Introduction

The cytosolic branched-chain aminotransferase (BCAT1) has gained significant interest regarding its role in the pathogenesis of haematological malignancies. Studies by Raffel et al. 2017 and Hattori et al. 2017 have demonstrated metabolic reprogramming mediated by BCAT1 in myeloid leukaemia cells, conferring a block in myeloid differentiation and oncogenic profile [1,2]. Canonically, BCAT1 catalyses the reversible transfer of nitrogen from the branched-chain amino acids (BCAA) leucine, isoleucine, and valine, to α-ketoglutarate, forming the respective branched-chain α-keto acid (BCKA) and glutamate [3]. Steady state kinetic studies indicate that under standard assay conditions (37 °C, pH 7.8) BCAT1 favours the reverse reaction, yielding BCAA and α-ketoglutarate as products, with *K_cat_*/*K_m_* ranging from 1473 × 10^3^ m^−1^s^−1^ to 4905 × 10^3^ m^−1^s^−1^ depending on the variable substrate [4]. Given the role that α-ketoglutarate and leucine play in cellular differentiation and autophagy [5,6,7], it is therefore unsurprising that BCAT1 mediated metabolic reprogramming in myeloid malignancy contributes to an oncogenic profile.

A key regulatory feature of BCAT1 is a redox-active CXXC motif situated ~10 Å from the active site [8]. Although the CXXC motif is conserved between BCAT1 and BCAT2 proteins, the latter of which is expressed in the mitochondria [9], these proteins differentially respond to reactive oxygen species (ROS) and reactive nitrogen species (RNS). Reversible oxidation of BCAT2 is achieved by <10 equivalents of hydrogen peroxide (H_2_O_2_), whereas >1000 H_2_O_2_ equivalents are required for BCAT1 oxidation [10]. The CXXC motif of both BCAT isoforms is modified by RNS, with BCAT1 showing differential sensitivity, depending on the RNS donor [11]. Both BCAT isoforms participate in *S*-glutathionylation under oxidative and nitrosative stress, which is reversible upon the coupled enzymatic action of the glutaredoxin and thioredoxin systems [10,11]. Redox chromatography and mass spectrometry revealed that the CXXC motif of both BCAT1 and BCAT2 facilitates redox-mediated interactions with numerous cellular proteins, indicating that these metabolic proteins have additional cellular roles [10,12]. One such role is where both isoforms were shown to have dithiol-disulphide isomerase activity suggesting a chaperone role in protein folding [13]. To understand the reductive capacity of BCAT, studies estimating the reduction mid-point potential (*E_m_*) suggest that BCAT1 has a greater reducing capacity (−260 mV) compared with BCAT2 (−220 mV) [14]. Taken together, these observations suggest a unique role for the BCAT1 CXXC motif and a potential reductive/antioxidant role that warrants investigation.

The role of ROS in myeloid leukaemia is well characterised [15,16]. Acute myeloid leukaemia (AML) is a rapidly progressing heterogeneous haematological malignancy defined by the clonal expansion of myeloid blast cells in the bone marrow [17]. If left untreated these blast cells will overcrowd, leading to bone marrow failure [17]. AML is the most common acute leukaemia to affect adults, with disease incidence increasing with age [18]. Long-term remission rates remain at ~50%, therefore investigating the molecular basis of this disease is paramount to establishing new treatment strategies [18]. Many studies have indicated a role for ROS in AML, which has been shown to potentiate oncogenic signalling [19,20,21]. Moreover, the expression of Nrf2 regulated antioxidant enzymes in AML such as glutaredoxin and thioredoxin reductase (TrxRD1) [22,23] has been shown to ‘buffer’ the AML against lethal levels of ROS, keeping the cell in a tumourigenic state that promotes cell proliferation, survival and may confer chemotherapy resistance [24,25].

BCAT1 expression level is shown to be raised in AML, and previous studies demonstrate metabolic adaptations as a result of canonical BCAT1 function [1,2]. However, the role of the BCAT1 CXXC motif has yet to be explored specifically in AML. We hypothesise that the BCAT1 CXXC motif may play a part in the redox homeostasis and redox-mediated cellular processes e.g., growth and survival. Therefore, the aim of this study was to evaluate the role of the BCAT1 CXXC motif in AML cells and understand how this may influence cellular redox processes.

## 2. Materials and Methods

### 2.1. Cell Lines

U937 (ATCC^®^ number CRL-1593.2^TM^) monocytic human myeloid leukaemia cell line and HEK293T (ATCC^®^ CRL-3216™) embryonic kidney cells for retroviral production were obtained from American Type Culture Collection (Glasgow, UK). U937 cells were maintained at 1 × 10^5^ and 2 × 10^6^ cells/mL, in RPMI-1640 media adjusted to contain 10% foetal calf serum (FCS) and 2 mM L-glutamine. HEK293T cells were seeded at 4 × 10^5^ cells/mL for 3-day subcultures, in DMEM adjusted to contain 10% FCS and 2 mM L-glutamine. Both cell lines were incubated at 37 °C and 5% CO_2_.

### 2.2. BCAT1 Site Directed Mutagenesis and Retroviral Transduction of U937 Cells

*BCAT1* cDNA transcript variant 5 (NM_001178094) contained in PCMV6-Entry vector (PS100001, Origene) was obtained from Origene (Rockville, MD, USA). Site-directed mutagenesis of the *BCAT1*-CXXC motif was performed using the Quikchange II Site-directed Mutagenesis Kit (Agilent, UK) and mutagenic primers (C338SF 5′-GGTACAGCCTGTGTTGTTAGCCCAGTTTCTGATATACTG-3′, C338SR 5′-CAGTATATCAGAAACTGGGCTAACAACACAGGCTGTACC-3′). Successful mutation was confirmed by capillary sequencing at the University of Birmingham DNA Sequencing Facility. *BCAT1-*wild type (WT) and CXXC motif mutant *BCAT1*-C338S were sub-cloned into a pLenti-C-Myc-DDK-IRES-Puro lentiviral gene expression vector (Origene, UK, see Appendix A) and Lentiviral particles containing pLenti-*BCAT1* vectors were generated using Lenti-vpak packaging system (Origene, MD, USA) in HEK293T cells. For lentiviral transduction 2 × 10^5^ cells were added to 1 mL of viral suspension and centrifuged at 900× *g* for 2 h at 32 °C. Stably transduced cells overexpressing *BCAT1* constructs were selected for incubation with complete RPMI supplemented with 0.5 µg/mL Puromycin.

### 2.3. Overexpression and Purification of Recombinant BCAT1 Protein in E. coli

Overexpression and purification of WT and CXXC motif mutant BCAT1 and BCAT2 protein was carried out in accordance with Davoodi et al., 1998 [3]. Briefly, WT and CXXC motif pLenti-*BCAT* was sub-cloned into pET28a vectors using the *Sal*I and *Nhe*I restriction sites. The pET28a-*BCAT* constructs were transformed into *E. coli* BL21DE3 competent cells and cultured to an optical density of 0.6–0.9, before inducing the over-expression of His-tagged BCAT1 protein with 1 mM isopropyl-β-D-thiogalactopyranoside. BCAT protein was purified from *E. coli* BL21 DE3 cell lysates using Ni-NTA agarose, in accordance with the manufacturer’s instructions (Qiagen, Manchester, UK). Ni-NTA purified BCAT protein was exchanged into 50 mM Tris (pH 7.5), 150 mM NaCl prior to proteolytic removal of the His-tag with 100 IU thrombin. As a final purification step, ion exchange chromatography was performed in 10 mM potassium phosphate, pH 8.0, on a HiTrap Q HP anion exchange chromatography column by means of a 0–100 mM NaCl gradient, which was set using an ÄKTA Pure with Fast Protein Liquid Chromatography Fraction Collector (GE Healthcare, Amersham, UK). Purification of BCAT protein was confirmed by SDS-PAGE and Coomassie blue staining. Spectrophotometric estimation of BCAT1 protein concentration was made at λ280 nm using the molar extinction coefficient of 86,300 M^−1^ cm^−1^ for the BCAT1 monomer and 80,000 M^−1^ cm^−1^ for the BCAT2 monomer [3].

### 2.4. Spectrophotometric Confirmation of Cysteine Mutation by DTNB Titration

To confirm site-directed mutagenesis of the BCAT1 CXXC motif at the protein level, spectrophotometric thiol group titration was performed using 5,5-dithio-bis-(2-nitrobenzoic acid) (DTNB) as previously described [3]. Briefly, 2 nmol of BCAT1 protein, previously exchanged into 50 mM Tris-HCl (pH 7.0), 1 mM EDTA, was titrated against 2 mM DTNB. The titration was monitored for 20 min at λ412 nm at room temperature in a spectrophotometer. The amount of free thiol group for WT and CXXC motif mutant BCAT1 protein was calculated using a molar extinction coefficient of 14,150 M^−1^ cm^−1^ for the nitro-5-thiobenzoate anion.

### 2.5. Hydrogen Peroxide Metabolism Assay

In accordance with the method developed by Li et al., 2007 [26], WT BCAT1, CXXC motif mutant BCAT1, and BCAT2 proteins were exchanged into 0.05 M phosphate buffer (pH 7.0), 10 mM sodium azide. BCAT protein was subsequently incubated with 5 mM hydrogen peroxide (H_2_O_2_) at room temperature in a quartz cuvette and monitored over time for the disappearance of H_2_O_2_ at λ240 nm in a UV spectrophotometer. 1 IU catalase was used as a positive control, minus the addition of 10 mM sodium azide. To confirm whether the disappearance of H_2_O_2_ observed by BCAT1 was Cys-thiol mediated, BCAT1 protein was incubated with 50× molar excess *N*-ethymaleimide prior to assay. DTNB titration was used to confirm the loss of BCAT1 Cys-thiols.

### 2.6. qPCR of BCAT1 and TrxRD1 in Transgenic U937 Cells

Puromycin-resistant U937 cell clones were assessed for expression of *BCAT1* at the transcript level by qPCR. RNA was extracted from transgenic U937 cells using Bioline’s isolate II RNA mini kit (BIO-52072) and cDNA was synthesised using a cDNA synthesis kit (PCR Biosystems, London, UK). cDNA samples were analysed for *BCAT1* and *Beta-2-Microglobulin* (*B2M*) expression using qPCR primers (BCAT1-Forward 5′-TGGAGAATGGTCCTAAGCTG-3′, BCAT1-Reverse 5′-GCACAATTGTCCAGTCGCTC 3′, B2M-Forward 5′-ATGAGTATGCCTGCCGTGTGA-3′, B2M-Reverse 5′-GGCATCTTCAAACCTCCATG-3′). The qPCR reaction was carried out using a Roche 480 LightCycler, cycling parameters were as follows: (1) Initial denaturation: 95 °C for 10 s, (2) Annealing: 63 °C for 10 s, (3) Extension 72 °C for 20 s for 35–40 cycles. For the *TrxRD1*, qPCR was performed in accordance with Karunanithi et al., 2021 [27] using the primers (TrxRD1-Forward 5′-AGACAGTTAAGCATGATTGG-3′ and TrxRD1-Reverse 5′-AATTGCCCATAAGCATTCTC-3′). Relative expression of *BCAT1* and *TrxRD1* to housekeeping gene *B2M* was calculated using the −2ΔΔ^CT^ method [28].

### 2.7. Western Blot Detection of Transgenic BCAT1 in U937 Cells

Transgenic BCAT1 proteins contain a C-terminal Myc-DDK FLAG tag (DYKDDDDK), therefore, to confirm BCAT1 overexpression in U937 cells 30 µg of whole cell lysate was separated by SDS-PAGE prior to detection with Anti-DDK (FLAG)-HRP mAb (Origene, London, UK). Expression was normalised to Cyclophilin-B loading control using 1:2000 Anti-Cyclophilin-B HRP (Abcam, Cambridge, UK).

### 2.8. In Vitro Growth Characterisation of Transgenic U937 Cells

Transduced U937 cell lines were cultured in T75 flasks seeded at 2 × 10^5^ cells/mL in a 10 mL total volume of complete RPMI (2 mM L-glutamine, 200 units/mL penicillin, 200 µg/mL streptomycin, and 10% *v*/*v* foetal calf serum) and incubated in a cell culture incubator (37 °C and 5% CO_2_) for 192 h. 100 µL samples were taken at 24 h intervals from 0–96 h and from 168–192 h. Cell density and viability were measured by flow cytometry using Viacount stain with Easyfit Cluster Analysis Algorithm (Merck-Millipore, London, UK). Exponential doubling time was calculated between 24–72 h. Cell cycle analysis of mid-exponential phase cells was performed by propidium iodide staining after 48 h. Cell apoptosis was measured by 7AAD (BioLegend, London, UK) and Annexin-V (BioLegend, London, UK) bivariate flow cytometric analysis.

### 2.9. Intracellular ROS and Glutathione Analysis of Transgenic U937 Cells

Intracellular ROS was measured by flow cytometry following 2′,7′-dichlorofluorescein diacetate (DCFDA) staining using a DCFDA Assay kit (Abcam, UK), in accordance with the manufacturer’s instructions. Oxidative stress was induced by serum deprivation i.e., incubation with complete RPMI minus foetal calf serum supplementation (0% FCS). Transduced U937 cells were incubated at a density of 5 × 10^5^ cells per well in a 96 well plate, in 100 µL complete RPMI ±10% FCS for 72 h, at 37 °C and 5% CO_2_. To control the contribution of cellular ROS, 10 mM *N*-acetyl-L-cysteine (NAC) (Sigma-Aldrich, Gillingham, UK) was added to the cultures prior to incubation. Following incubation, total cellular glutathione was measured by spectrophotometry using a 96 well assay kit (Sigma-Aldrich, Gillingham, UK). Oxidised glutathione (GSSG) was evaluated by incubating the transgenic U937 cells with 50 mM NEM in PBS prior to carrying out the assay and subtracting from the respective total glutathione measurement. Cellular Redox potential (*E_h_*) was determined by the ratio of reduced/oxidised glutathione (GSH/GSSG) content calculated from using the Nernst Equation: *E_h_* = *E_m_* − (RT/nF)2.303 log [2GSH]/[GSSG], as described in Coles et al., 2012 [14].

### 2.10. Evaluation Transgenic U937 Cellular Differentiation

To differentiate transgenic U937 cells to monocyte/macrophage, 1 × 10^6^ cells/mL were incubated with complete RPMI supplemented with a final concentration of 10 nM phorbol 12-myristate 13-acetate (PMA) (Sigma-Aldrich, Gillingham, UK) in 24 well plates for 48 h. To control the metabolic and redox contribution of BCAT1, the media was supplemented with 20 mM Gabapentin (Sigma-Aldrich, Gillingham, UK) and/or 10 mM NAC respectively. Intracellular ROS ± PMA was measured by DCFDA staining as previously described. To examine cellular differentiation, U937 cells were incubated with anti-CD14 FITC (Biolegend), anti-CD68 PE (Biolegend), anti-CD36 PE (Biolegend, London, UK) and anti-CD11b FITC (Biolegend, London, UK) prior to flow cytometric analysis, where % positive cells and median fluorescence intensity (MFI) were evaluated. PMA treated cells were stained with Giemsa and imaged via light microscopy to evaluate macrophage morphology.

### 2.11. Data Analysis

Statistical significance was assessed by 2-way ANOVA and Bonferroni post-test for multiple comparisons, where a *p* < 0.05 was considered statistically significant. All data were analysed by GraphPad Prism v5.1.3.

## 3. Results

### 3.1. Purified WT BCAT1 Protein Metabolises Hydrogen Peroxide In Vitro

Wild-type (WT) and CXXC motif mutant (SXXS) recombinant BCAT1 proteins (NM_001178094) were generated in *E. coli* BL21DE3 cells as previously described [3]. The CXXC motif was mutagenised in the pCMV6-cDNA-Entry Vector (Origene) and sequenced, prior to sub-cloning into the pET28a protein expression vector. The data presented in Figure 1a–c confirm successful mutagenesis and sub-cloning of the respective *BCAT1* genes. To validate successful CXXC motif mutagenesis at the protein level, WT BCAT1 and SXXS BCAT1 proteins were purified by ion-exchange chromatography as illustrated in Figure 1d–f, prior to DNTB titration, which showed two fewer Cys residues for the SXXS BCAT1, compared with WT BACT1, Figure 1g.

To evaluate the reductive potential and potential antioxidant role of the BCAT1 CXXC motif, purified recombinant BCAT1 protein was incubated with 5 mM H_2_O_2_, and monitored at λ240 nm. The data presented in Figure 2a shows the absorbance change at λ240 nm, signifying the disappearance of H_2_O_2_. The data illustrate that 1 IU catalase was the most efficient at catalysing the removal of H_2_O_2_, followed by WT BCAT1 where the CXXC motif is unmated. WT BCAT2 or the phosphate buffer control (data not shown) had no effect over the 5-min period evaluated. To confirm that the effect observed by BCAT1 was CXXC motif mediated, SXXS BCAT1 was subsequently evaluated. Data presented in Figure 2b,c demonstrate that the SXXS BCAT1 protein lost the capacity to catalyse the removal of H_2_O_2_. To further confirm that this observation is Cys-thiol mediated, WT BCAT1 (CXXC BCAT1) was pre-treated with 50 molar excess NEM, a thiol-reactive alkylating agent. DTNB titration analysis confirmed that there were 0 titratable free thiol groups following NEM treatment, Figure 2b. The data shown in Figure 2c demonstrates that, like SXXS BCAT1, the NEM treated CXXC BCAT1 lost the capacity to catalyse the removal of H_2_O_2_. To establish whether BCAT1 has oxidoreductase activity, NADPH was added to the assay. The data presented in Figure 2d show the oxidation of NADPH as monitored by the Δλ340 nm. Taken together, these findings suggest that the BCAT1 CXXC motif has the capacity to reduce H_2_O_2_ in in vitro assays and may have an antioxidant role.

### 3.2. Overexpression of BCAT1 in U937 Cells Increases Growth and Survival and Is Associated with a More Reductive Redox Environment

Next, we evaluated whether the BCAT1 CXXC motif could provide antioxidant protection in AML cells. Hence the effect of the BCAT1 CXXC motif with respect to the redox environment and redox-mediated processes in AML, including cell growth and survival [15,19,20] was investigated. Appendix A shows the OriGene pLenti-cMyc-DDK-IRES-Puro vector map. The *WT BCAT1* and *CXXS BCAT1* genes were subcloned into the *Sgf*I and *Mlu*I restriction site. Figure 3a–e confirms successful subcloning and the generation of transduced U937 cells, two of which stably overexpress WT BCAT1 and CXXS BCAT1 protein, as confirmed by Western-blotting, Figure 3f. All three transduced cells lines are resistant to puromycin with respect to non-transfected U937 controls. Gene expression analyses by qPCR demonstrated a ~6-fold increase in both *WT* and *CXXS BCAT1* mRNA transcripts with respect U937 transgenic cells.

Growth kinetics and maximum cell densities for all 3 transgenic U937 cells were initially characterised. The data presented in Figure 4 illustrate that WT BCAT1 and CXXS BCAT1 U937 cells reached significantly higher cell densities (1.57 × 10^6^ ± 8.53 × 10^4^ cells/mL, *p* < 0.001 and 1.51 × 10^6^ ± 8.16 × 10^4^ cells/mL, *p* < 0.01) at the end of the exponential growth compared to Vector controls (1.01 × 10^6^ ± 7.71 × 10^4^ cells/mL). The data presented in Figure 4 also show that BCAT1 overexpressing cells display consistently increased viability compared to control during the stationary and death phase, importantly WT BCAT1 cells are significantly more viable compared to CXXS BCAT1 after 192 h (69.1 ± 5.5% vs. 57.4 ± 4.8, *p* < 0.05) indicating the CXXC motif may have a protective effect during the death phase. Vector control cells at 192 h displayed the least viability at 32.6 ± 3.17%, which was significantly lower than both WT BCAT1 and CXXS BCAT1 over-expressing cells (*p* < 0.001).

Next, the intracellular redox environment was assessed by measurement of ROS, total glutathione (GSH) content, and redox potential (*E_h_*), at exponential (48 h), stationary (96 h), and death phase (168 h). It was important to evaluate GSH and *E_h_* in particular, given the recent observation that BCAT1 expression alters GSH/GSSG in macrophages [29]. Figure 4e shows that intracellular ROS levels as assessed by DCFDA were significantly lower at 96 h and 168 h for WT BCAT1 cells compared with Vector controls (*p* < 0.001), as was the case for CXXS BCAT1 cells compared with Vector controls (*p* < 0.01). Notably, this data corresponded with decreased intracellular *E_h_* for both WT BCAT1 and CXXS BCAT1 cells at 96 (*p* < 0.05) and 168 h (*p* < 0.01) compared with vector controls, indicating a more ‘reducing’ intracellular environment for BCAT1 expressing cells. At 96 h, total glutathione content was significantly increased in both WT BCAT1 (*p* < 0.01) and CXXS BCAT1 cells (*p* < 0.05) compared with vector controls (Figure 4g). No significant differences were observed at 48 and 168 h. The data also illustrate that WT BCAT1 cells were significantly more viable at 168 h (*p* < 0.001), compared with Vector controls or CXXS BCAT1 (4 h).

### 3.3. BCAT1 CXXC Motif Provides Protection against Serum Deprivation Induced Oxidative Stress

To observe the potential antioxidant effect of the CXXC motif, cell lines were placed under oxidative stress elicited by serum deprivation [30]. Figure 5 illustrates that serum deprivation resulted in an increase in the DCFDA MFI across all cells lines signifying higher ROS.

However, WT BCAT1 cells displayed a significantly lower overall DCFDA MFI (1533 ± 51) compared to either CXXS BCAT1 (2554 ± 93; *p* < 0.001) or Vector control cells (3598 ± 154; *p* < 0.001), revealing the extent to which the CXXC motif mediates a reduction in intracellular ROS following serum deprivation. The addition of 10 mM NAC abrogated this effect, as shown in Figure 5a. The effect of these treatments on cell viability and cell density illustrates that BCAT1 overexpressing cells consistently display increased viability during serum deprivation (*p* < 0.001), with WT BCAT1 cells displaying significantly more viability compared with CXXS BCAT1 cells (*p* < 0.05), as shown in Figure 5b, with a similar trend observed for cell density (Figure 5c). Analysis of fold increase in DCFDA MFI reveals that intracellular ROS increases in a CXXC dependent manner, where CXXS BCAT1 expressing cells displayed an 80.41 ± 6.57-fold increase in DCFDA MFI compared with a 43.61 ± 3.23-fold increase for WT BCAT1, Figure 5d (*p* < 0.001).

Next, the effect of serum deprivation on cell viability and apoptosis was measured by 7AAD/Annexin-V(APC) flow cytometry analysis. The data shown in Figure 5e,f demonstrate that serum deprivation significantly increases the percentage of late apoptotic cells (7AAD^+^/Annexin-V^+^) cells, across all cell lines. However, WT BCAT1 cells display a substantially lower proportion of late apoptotic cells (22.9 ± 4.4%) compared to either CXXS BCAT1 cells (38.3 ± 4.7%; *p* < 0.01) or Vector controls (74.6 ± 2.1%; *p* < 0.001), indicating that the CXXC motif partially protects U937 cells under serum-starved conditions. Notably, the addition of 10 mM NAC significantly reduced the late apoptotic population across all cells lines and abrogated the difference between WT BCAT1 and CXXS BCAT1 cells, mirroring the effect observed with DCFDA MFI following NAC treatment.

### 3.4. The Effect of the BCAT1 Inhibitor, Gabapentin, on BCAT1-WT Overexpressing U937 Cells

The previous data sets illustrated that WT BCAT1 U937 cells were more resistant to ROS and that total glutathione levels between WT BCAT cells and CXXS BCAT1 cells were comparable, suggesting that the CXXC motif contributes in some way to these observations. Next, we wanted to evaluate the effects of the BCAT1 CXXC motif on ROS-mediated processes, such as cellular differentiation. Thus, the minimum inhibitory concentration (X0) of the BCAT1 competitive inhibitor Gabapentin [8] and the dose required to inhibit 50% cells (IC_50_), was determined for Vector control, WT BACT1 and CXXS BCAT1 overexpressing U937 cells. Figure 6a shows the Gabapentin dose-response curves for each cell line as calculated by evaluating cell viability (%). Figure 6b shows the Gabapentin dose-response curves for each cell line as calculated by evaluating cell density. For both methods, it appears that WT BCAT1 overexpressing cells display slightly more sensitivity to Gabapentin, compared with Vector controls and CXXS BCAT1 cells. To investigate this further, the average IC_50_ and X0 concentrations with respect to Gabapentin, for each cell line were compared. Figure 6c illustrates that WT BCAT1 cells were significantly more sensitive to the effects of Gabapentin (IC_50_ = 33.92 ± 0.22 mM and X0 = 24.38 ± 0.72 mM), as determined by cell viability, compared with Vector controls or CXXS BCAT1 cells (*p* < 0.05). The trend was similar for calculations based on cell density as illustrated in Figure 6d, where the IC_50_ and X0 for WT BCAT1 cells were 28.67 ± 1.07 mM and 18.35 ± 1.29 mM respectively. These data along with the non-linear regression analysis are summarised in Table 1. 

### 3.5. BCAT1 CXXC Motif Modulates Intracellular ROS and Altered Myeloid Phenotypic Expression in Response to PMA Induced Differentiation

Since ROS are a central component in monocyte to macrophage differentiation [21,31], the role of the BCAT1 CXXC motif in this process was next examined. U937 cells will differentiate in response to treatment with PMA, which increases cellular ROS [32]. The data presented in Figure 7a show that PMA treatment resulted in an increased DCFDA MFI signal across all cell lines, with WT BCAT1 cells displaying significantly lower DCFDA MFI (36.9 ± 1.2) than either CXXS BCAT1 cells (46.9 ± 2.6; *p* < 0.05) or Vector control cells (56.0 ± 4.3; *p* < 0.001). Cell morphology following PMA is presented in Figure 7b. Following treatment with PMA [or] PMA+20 mM Gabapentin (BCAT1 inhibitor), the percentage of CD14-positive cells (CD14^+^) was significantly lower for WT BCAT1 compared with Vector controls (*p* < 0.01 and *p* < 0.001) respectively. The addition of NAC was able to partially restore the phenotype, illustrating a ROS role. A similar trend was observed for CD68^+^ cells, where WT BCAT1 was significantly lower for PMA treated compared with Vector controls (*p* < 0.001) and CXXS BCAT1 cells (*p* < 0.01). A consistent trend was observed following PMA+20 mM Gabapentin treatment for CD68 expression.

Figure 8a shows that for CD11b, the effects were less significant, however PMA mediated an increase in CD11b^+^ cells across all cell lines, with WT BCAT1 cells (30.9 ± 14.4%; *p* < 0.05) and CXXS BCAT1 cells (35.0 ± 19.4%; *p* < 0.05) significantly decreased compared to Vector controls (62.6 ± 29.6%). Treatment with 20 mM Gabapentin prior to PMA displayed a lower percentage of CD11b^+^ cells for WT BCAT1 (26.7 ± 1.5%), compared with CXXS BCAT1 (38.4 ± 4.7%) and Vector controls (51.9 ± 4.5%), however these data did not reach significance. The addition of NAC was able to restore the basal phenotype for CD11b, demonstrating that PMA induced CD11b^+^ expression is also redox regulated.

Interestingly, for CD36 the percentage of positive cells was significantly lower for WT BCAT1 at baseline, compared with Vector controls (*p* < 0.001) and CXXS BCAT1 cells (*p* < 0.01). Figure 8b shows that the addition of the antioxidant NAC was able to rescue the basal phenotype for CD36 expression, with the percentage of CD36^+^ WT BCAT1 significantly lower (35.6 ± 1.5%; *p* < 0.001) demonstrating PMA induced CD36 expression is redox regulated. Representative flow cytometric bivariate analysis plot data for CD36/CD11b expression are shown in Figure 8c. Taken together with Figure 7, these data illustrate a role for the BCAT1 CXXC in U937 cell differentiation.

### 3.6. BCAT1 CXXC Motif Is Associated with Altered Expression of Nrf2 Regulated Genes in U937 Cells

The regulation of CD36 expression has been shown to be mediated by Nrf2 [33], suggesting that the BCAT1 CXXC motif may impact Nrf2 regulated genes. To further investigate this, we examined the influence of the BCAT1 CXXC motif on the expression *thioredoxin reductase 1* (*TrxRD1*), which is also regulated by Nrf2 [34]. The data presented in Figure 9 compares *TrxRD1* gene expression levels between WT BCAT1, CXXS BCAT1, and Vector control U937 cells. As illustrated, there is significantly lower relative gene expression of *TrxRD1* for both CXXS BCAT1 (2.92 ± 0.16, *p* < 0.01) and Vector controls (3.70 ± 0.40, *p* < 0.001) compared with WT BCAT1 U937 cells (1.02 ± 0.21).

## 4. Discussion

This study set out to evaluate the role of the BCAT1 CXXC motif in AML and to understand how this may influence cellular redox processes. Based on previous studies [9,10,11,14], we hypothesised a unique ‘redox homeostasis’ role for the BCAT1 CXXC motif, which is based on the differential sensitivity of the BCAT1 CXXC motif to ROS as well as the more reductive *E_h_*. To investigate this, *BCAT1* cDNA (NM_001178094) was mutagenized in a way to alter the codons at positions 335 and 338, which relates to the CXXC motif [10], creating constructs that will produce WT BCAT1 (CXXC motif intact) and SXXS BCAT1. These mutants have been previously documented and characterised [10,11,14]. To confirm the successful mutation at the protein level, DTNB titration illustrated two fewer Cys-thiols for SXXS BCAT compared with WT BCAT (Figure 1), conducive to previous findings [10,11].

Initially, we investigated the antioxidant potential of the BCAT1 protein biochemically (Figure 2). WT BCAT1 demonstrated a novel antioxidant ‘peroxidase-like’ activity, which was not present for SXXS BCAT1, NEM pre-treated BCAT1, or WT BCAT2. Both BCAT1 and BCAT2 belong to the fold type-IV PLP dependent aminotransferase family [35]. Interestingly, the cladogram presented in Appendix A shows that the BCAT proteins appear to have more in common with Cys antioxidant proteins e.g., the peroxiredoxin (Prdx) family [36], than the broader PLP dependent aminotransferases. However, the BCAT proteins lack the archetypal PxxxT/SxxC ‘peroxiredoxin’ motif [37], therefore the antioxidant peroxidase-like activity as observed in Figure 2 could be explained by a dithiol-disulphide exchange mechanism, which was previously noted for both BCAT proteins in a chaperone capacity [13] and is more common with the oxidoreductases [38].

We next determined whether BCAT1 displayed evidence of pyridine disulphide oxidoreductase activity by utilising reducing equivalents from NADPH [38]. Figure 2 illustrates the oxidation of NADPH in our H_2_O_2_ assay, suggesting that BCAT1 may be coupled to NADPH under oxidising conditions, similar to the thioredoxin/thioredoxin reductase (Trx/TrxRD1) system [38,39]. Previous data show interaction with the Trx/TrxRD1 antioxidant system and BCAT1 [10]. Moreover, the BCAT1 CXXC motif can form disulphide interactions with numerous proteins under oxidative stress [10], similar to Trx [39]. Thus, taken together, it is conceivable that BCAT1 may function as a disulphide oxidoreductase under certain biochemical parameters. To the best of our knowledge, this has not been demonstrated before.

Given the potential antioxidant capacity of the BCAT1 CXXC motif, it was next important to evaluate this in an AML cell model. Previous studies show that the U937 immortalised AML cell line is a good model to study myeloid malignancy and redox-mediated myeloid cellular processes in vitro [32,40,41]. The U937 cell line was originally isolated from the pleural effusion of a patient with histiocytic lymphoma [42]. Early characterisation studies showed that U937 cells differed from typical lymphoblastoid cell lines [42]. Because of their pro-monocytic phenotype, many regard the U937 cell line to be representative of an AML subtype M5, under the French-American-British (FAB) classification system [43,44,45,46]. This pro-monocytic phenotype and cytochemistry, accounted for the wide use of U937 cells as a model to study AML, including ROS mediated monocyte to macrophage transition [32,47,48]. We identified U937 cells as having a relatively low basal expression of BCAT1 compared to other myeloid cell lines [49], which would allow an effective comparison between BCAT1 overexpressing cells to the vector control. Taken together, the U937 cell line presented as a good cellular model to study the putative antioxidant effect of the BCAT1 CXXC motif in myeloid differentiation, which is a defining feature of AML [50]. However, our study is limited to a single FAB subtype model. Future work could extend our analysis and examine the antioxidant capacity of the BCAT1 CXXC motif in other FAB subtype models, for example HL60 (AML, M2) and NB4 (AML M3 aka APL) [45].

Several transduced U937 cell lines were generated, that stably overexpress WT BCAT1 protein or CXXS BCAT1 protein. Empty Vector U937 cells were also generated to control the puromycin resistance (Figure 3). A CXXS BCAT1 mutant, which preserves the N-terminal Cys was selected over SXXS BCAT1, since previous data demonstrate that under physiological reduced conditions, CXXS BCAT1 retains significantly more aminotransferase activity, compared with SXXS BCAT [10]. Plus, under cellular reduced/oxidised glutathione ratios (GSH/GSSG), studies show that the majority of the N-terminal Cys (C335) is largely S-glutathionylated, which protects BCAT1 enzymatic turnover [10]. In this way, CXXC motif mediated antioxidant processes could be better distinguished from the BCAT1 aminotransferase activity.

WT BCAT1 and CXXS BCAT1 U937 cells displayed ~6-fold increase in BCAT1 expression over Empty Vector Controls, which is comparable to the ‘high’ BCAT1 expression levels observed in primary AML patients [2]. The data presented in Figure 4 show that both WT BCAT1 and CXXS BCAT1 U937 cells reach significantly higher cell densities and demonstrate increased survival under standard cell culture conditions, compared with Empty Vectors. This data is likely explained by altered BCAA metabolism. Previous biochemical studies assessing in vitro enzyme kinetics, demonstrate that BCAT1 metabolism favours the production alpha-ketoglutarate (α-KG) and leucine [3], however current literature indicate that the direction of this reaction in vivo is context dependant. In myeloid leukaemia, rapidly dividing blast crisis cells favour this reaction direction, whilst more quiescent leukemic stem cell populations have been found to favour the reverse direction i.e., the production of glutamate and BCKA [1,2]. In proliferating cells, α-KG anaplerotically replenishes the TCA cycle, whilst leucine activates mTOR stimulating growth and survival [51,52], which may explain the greater cell densities and survival as seen in Figure 4.

Differences between WT BCAT1 and CXXS BCAT1 U937 cells were noted at time points >150 h (Figure 4). Both cell proliferation and cell survival are ROS-mediated processes [15,20,53], therefore as cells progress in culture, the BCAT1 CXXC motif may act as a redox buffer prolonging proliferation and enhancing survival. To investigate this further, cellular ROS levels were evaluated using a H_2_O_2_ reactive probe, DCFDA [54]. Figure 4 illustrates significantly less DCFDA signal for both WT BCAT1 and CXXS BCAT1 cells compared with Empty Vector Controls under standard cell culture conditions (10% FCS). This suggests that overall, BCAT1 aminotransferase activity mediates a lower ROS environment in standard culture conditions. Previous works demonstrate altered cellular [GSH] in the context of BCAT1 activity in macrophages [29]. Thus, during the stationary (96 h) phase and death phase (168 h), BCAT1 overexpressing U937 cells may favour glutamate production, mediating increased GSH biosynthesis [1,2]. Support for this notion comes from studies that demonstrate that, during the stationary phase, exhaustion of glutamine would lead to a rapid decrease in the availability of glutamate, shifting the reaction equilibrium of BCAT1 in favour of glutamate synthesis and GSH production [55]. This increase in GSH seen for WT BCAT1 and CXXS BCAT1 U937 cells (Figure 4), may in turn be used by cellular antioxidant systems i.e., Glutathione peroxidase (GPx) [56]. This theory explains the reduction in DCFDA signal for BCAT1 transformed cells [54]. To corroborate these findings, cellular redox potential as generated from the Nernst Equation using the GSH/GSSG redox couple [57], illustrates that both WT BCAT1 and CXXS BCAT1 U937 cells have significantly more ‘negative’ cell redox potentials compared with Empty Vector controls, signifying a more ‘reductive’ cellular environment [57]. Taken together with the novel antioxidant role, BCAT1 may be mediating a previously uncharacterised ‘reductive stress’ in AML cells [58].

It was next important to evaluate the effect of the BCAT1 CXXC motif under cell culture conditions that impose oxidative stress. For these studies, cells were cultured under serum-starved conditions (0% FCS) for 72 h, which generates endogenous ROS [30]. Here, WT BCAT1 cells displayed significantly lower levels of ROS compared with CXXS BCAT1 or Empty Vector, with no detectable difference in ROS fold-change between CXXS BCAT1 and Empty Vector cells (Figure 5). Taken together, with the biochemical data presented in Figure 2, these findings provide evidence of a novel antioxidant role for the BCAT1 CXXC motif in AML cells under oxidative stress conditions. The addition of the widely used antioxidant, N-acetyl cysteine (NAC), was able to restore ROS to equivalent levels across all U937 transformed cell lines, demonstrating the importance of Cys in cellular redox homeostasis. This finding was irrespective of BCAT1 expression level or CXXC motif mutational status (Figure 5). When examining the effect of serum deprivation on apoptosis, the data show a similar trend, with the BCAT1 CXXC motif cells displaying significantly more cell survival (Figure 5). The effect of CXXC motifs in ROS-mediated cell death has been examined for the widely studied antioxidant protein thioredoxin (Trx). It is demonstrated that the Trx CXXC motif forms a catalytic reaction centre, which acts as a protein disulphide oxidoreductase in reducing intermolecular and intramolecular disulphide bonds of oxidised proteins [59]. Trx during serum deprivation in the monocytic AML cell line, THP1, has been shown to protect cells from apoptosis, linking to reduced cellular levels of ROS [30]. Whilst the antioxidant effects of the Trx system have been studied, the potential antioxidant properties of CXXC motifs in proteins outside of this system are less well understood. Recently, Cys to Ser site-directed mutagenesis of a CXXC motif in Mesencephalic Astrocyte-derived Neurotrophic Factor (MANF) demonstrated that the MANF CXXC motif provides protection from oxidative stress-induced apoptosis and reduces intracellular ROS [60]. This finding is similar to the data presented here and further supports an antioxidant role for the BCAT1 CXXC motif.

To better distinguish the contribution of the BCAT CXXC motif from the metabolic function, it was important to inhibit BCAT1 aminotransferase activity in the transformed U937 cells. To this end, dose responses of the BCAT1 competitive inhibitor ‘Gabapentin’ [8] were performed. The data presented in Figure 6 show the dose-dependent effect of Gabapentin on the growth and viability of WT BCAT1 U937 cells. The IC_50_ ranged between 28.78 mM and 33.97 mM concordantly. In a previous study, 10 mM Gabapentin was found to inhibit the growth of the HCT116 colorectal cancer cell line, for which BCAT1 expression is naturally low [52]. However, our data show the IC_50_ for Gabapentin is over twice as concentrated. This finding is unsurprising, given that BCAT1 expression levels are ~6-fold greater than baseline in our transformed cells (Figure 3). For subsequent studies, it was important to select a concentration of Gabapentin that would inhibit sufficient BCAT1 aminotransferase activity in the U937 cells, without having a major influence on cell growth and survival. To this end, 20 mM Gabapentin was selected as a concentration that has minimum impact on cell viability and cell density (Figure 6).

ROS are central mediators in the differentiation of myeloid cells, including U397 cells [32,40]. Moreover, ROS are implicated in AML pathogenesis, a disease that is defined by a developmental block in myeloid cell differentiation [15,53,61]. Since U937 cells follow a macrophage lineage [46,47], we next evaluated whether the BCAT1 CXXC motif could mitigate myeloid differentiation, by examining; CD11b CD14, CD68, and CD36 expression [21,32,33]. To control for BCAT1 aminotransferase metabolic activity, assays were performed +/− 20 mM Gabapentin. Phorbol esters, such as PMA are used widely for the differentiation of myeloid cells, which function in part by increasing intracellular ROS [32]. The data presented in Figure 7 and Figure 8 show that PMA treatment was linked with an overall increase in cellular ROS, with WT BCAT1 cells displaying significantly lower ROS following PMA treatment. This finding agrees with previous work linking ROS to PMA treatment [32] and further supports a novel antioxidant role for the BCAT1 CXXC motif in our study.

Next the effect of PMA treatment on the differentiation markers; CD11b, CD14, CD68, and CD36 was investigated across all transformed U937 cells. Figure 7 and Figure 8 show that the basal level of CD11b^+^, CD14^+^, CD68^+,^ and CD36^+^ cells was significantly lower for WT BCAT1 and CXXS BCAT1 cells. Following 20 mM Gabapentin pre-treatment, CD14, CD68, and CD36 expression was significantly lower for WT BCAT cells, suggesting a specific role for the BCAT1 CXXC motif in this process. This is the first study of its kind to demonstrate the BCAT1 CXXC motif mediated regulation of myeloid linage markers and supports the notion that the BCAT1 CXXC motif may be implicated in the pathogenesis of AML, which is characterized by a block in myeloid differentiation [17]. Most strikingly, the percentage of CD36^+^ cells was reduced significantly (~50%) for WT BCAT1 following NAC pretreatment (Figure 8). NAC is a Cys analogue and serves as a substrate for the regeneration of cellular GSH [62]. This is important for CXXC motif mediated antioxidant systems, where the addition of NAC and regeneration of cellular GSH has been shown to increase the activity of GPx [62,63].

It is previously demonstrated that the BCAT1 CXXC motif has a similar biochemical property, utilising GSH reducing power to maintain the CXXC motif via an S-glutathionylated adduct [10,14]. Thus, under PMA mediated oxidative stress following NAC pre-treatment, the N-terminal Cys of the CXXS BCAT1 mutant is likely to become S-glutathionylated, preserving BCAT1 aminotransferase activity [10], however, the potential antioxidant activity is now abolished since redox cycling can no longer function [64]. Whilst the cellular pool of NAC and GSH may increase in the context of our assays, for the CXXS BACT1 cells it may be that these redox couples become progressively oxidised through the PMA treatment process.

Previously, CD36 has been found to be regulated, in part, by the antioxidant transcription factor Nrf2. Under conditions of oxidative stress, Nrf2 initiates the transcription of numerous antioxidant response genes [34]. Since CD36 may be Nrf2 regulated, it would be expected that the percent of CD36^+^ cells would increase under oxidative stress. The data presented here support this hypothesis, showing that the ROS levels are higher in PMA treated CXXS BCAT1 cells, which in turn have increased CD36^+^ expression (Figure 8). Furthermore, gene expression data analysed from BloodSpot [65] shows that high *BCAT1* gene expression correlates with lower *CD36* gene expression in certain AML patients (Appendix A). Thus, to further investigate a link between the BCAT1 CXXC motif and Nrf2 regulated gene expression, we evaluated the Nrf2 regulated gene, thioredoxin reductase-1 (*TrxRD1*) [34] in our transformed U937 cells.

The data presented in Figure 9 show a significant reduction in *TrxRD1* gene expression for WT BCAT1 U937 cells, compared with CXXS BCAT1 or Empty Vector controls. Taken with the CD36 expression data (Figure 8), this finding suggests that there may be less Nrf2 activation in the WT BCAT1 cells, implicating the BCAT1 CXXC motif more widely with the antioxidant signalling network. This notion is supported by previous studies illustrating that CXXC motif antioxidant proteins can modulate the activity of Nrf2 [66,67]. The data presented here demonstrate less ROS and less Nrf2 regulated gene expression for WT BCAT1 U937 cells, compared with CXXS BCAT1 cells, further supporting a novel antioxidant role for the BCAT1 CXXC motif.

## 5. Conclusions

This study set out to evaluate the role of the BCAT1 CXXC motif in AML cells, and to understand how this may influence cellular redox processes important for oncogenesis. The biochemical data presented suggest that the BCAT1 CXXC motif can reduce H_2_O_2_ and utilise reducing equivalents from NADPH in this process. In U937 cells, the BCAT1 CXXC motif is significantly linked to low levels of cellular ROS. In WT BCAT1 cells, there is overall survival and proliferative advantage compared with CXXS cells. Both cell survival and proliferation are Hallmarks of Cancer and may be mediated by cellular ROS. Under serum-starved conditions, ROS levels for WT U937 cells are significantly buffered, compared with CXXS of Vector Control cells. There is also significantly less PMA mediated cellular ROS for WT BCAT1 cells and correspondingly less macrophage differentiation, as determined by CD11b, CD14, CD68, and CD36. Finally, the data support that the lower levels of ROS may influence Nrf2 regulated gene transcription. Taken together, this study presents the first evidence to suggest that the BCAT1 CXXC has a novel antioxidant role, which may play a role in AML pathogenesis.

## Figures and Tables

**Figure 1 antioxidants-11-00683-f001:**
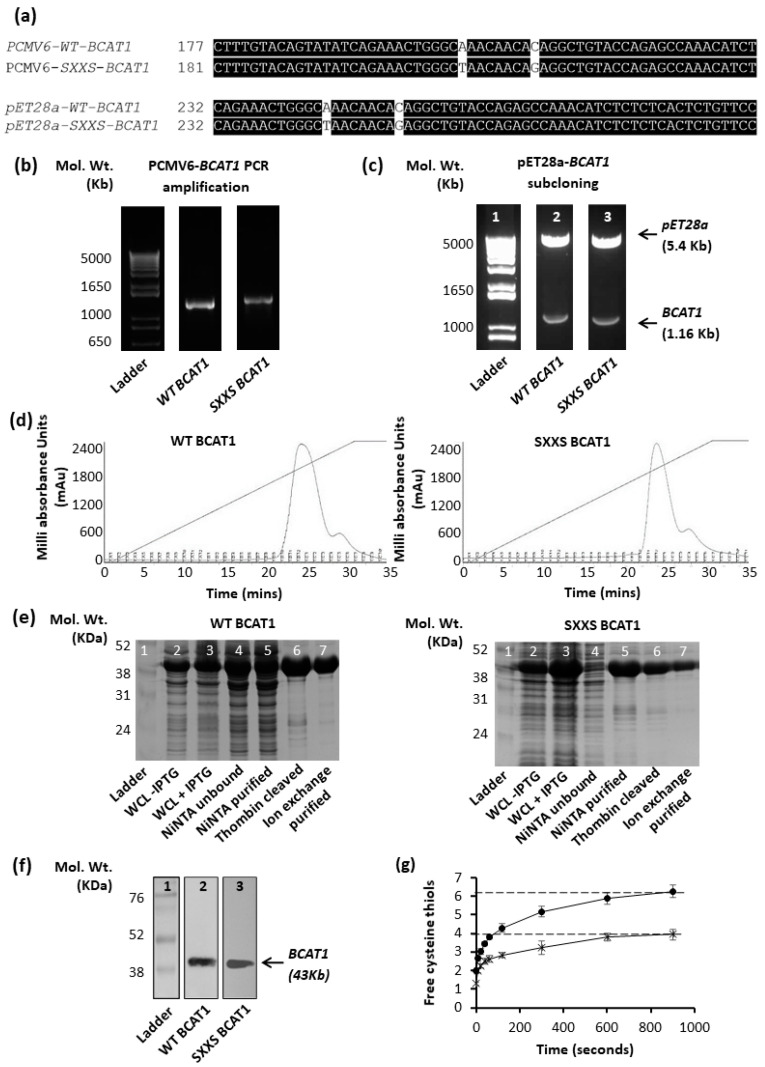
Site directed mutagenesis, pET28a, subcloning and over-expression/purification of WT and SXXS BCAT1 proteins. (**a**) Sequence alignment of PCMV6-*BCAT1* and subcloned pET28a-*BCAT1* using M-Coffee (OmicX, France), CXXC motif base substitutions are highlighted in white, confirming site-directed mutagenesis was present after subcloning and capillary sequencing. (**b**) Agarose gel image displaying electrophoretic separation of PCR amplified *WT BCAT1* and mutagenised *SXXS BCAT1* prior to ligation into pET28a vector. (**c**) Gel image showing endonuclease digested pET28-*BCAT1* separated by agarose gel electrophoresis, presence of a 1.16 Kb band confirms successful insertion of *BCAT1*. (**d**) Chromatogram displaying absorbance of WT BCAT1 protein and SXXS BCAT1 protein eluted from the strong anion columns (HiTrapTM Q HP), using the ÄKTA pure protein purification system. Samples were eluted by 0–0.5 M NaCl concentration gradient and fractions were pooled for analysis. (**e**) SDS-PAGE displaying WT BCAT1 and SXXS BCAT1 purified protein fractions at (1) uninduced, (2) induced, (3) Ni-NTA, (4) flow-through, (5) thrombin restriction, (6) gel filtration and (7) Ion-exchange chromatography. (**f**) Western blot detection of purified recombinant BCAT1 protein, using anti-ECA39[BCAT1] from Santa Cruz (sc-517185). (**g**) DTNB titration of BCAT1 protein displaying the number of titratable Cys-thiols for WT BCAT1 (circles) and SXXS BCAT1 (crosses) (*n* = 3, data presented are mean ± standard deviation).

**Figure 2 antioxidants-11-00683-f002:**
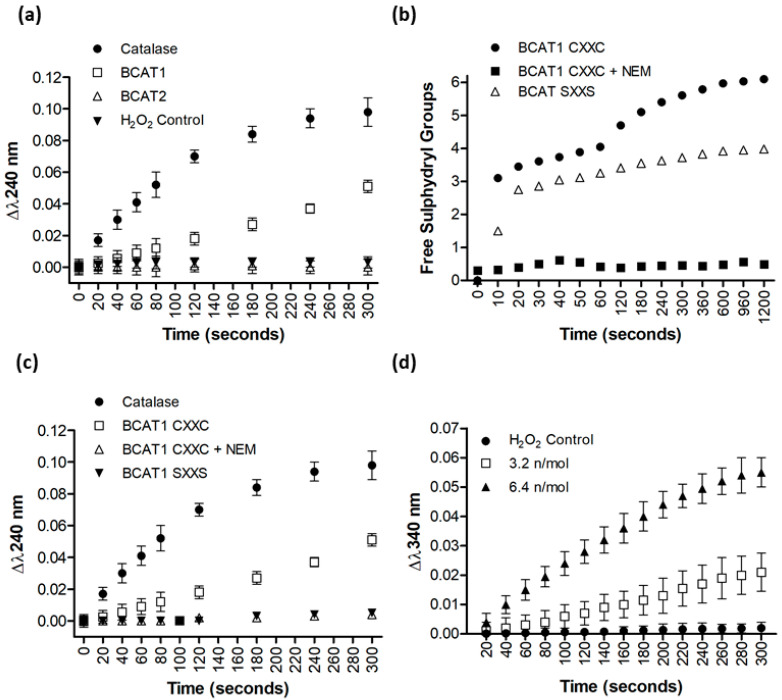
Purified recombinant WT BCAT1 protein metabolises H_2_O_2_ in vitro. (**a**) Evaluation of the capacity of WT BCAT1 and WT BCAT2 to metabolise H_2_O_2_ compared with catalase. (**b**) Site-directed mutagenesis of the BCAT1 CXXC motif was confirmed by DTNB titration for the number of free cysteine thiol for WT BCAT1, WT BCAT1 treated with NEM and CXXC motif mutant BCAT1 (SXXS). to metabolise H_2_O_2_ compared with catalase. (**c**) Evaluation of the capacity of WT BCAT1, WT BCAT1 treated with NEM and SXXS BCAT1 to metabolise H_2_O_2_ compared with catalase. (**d**) Evaluation of the capacity of BCAT1 to utilise NADPH in the metabolism of H_2_O_2_. Data presented are mean ± SD, *n* = 3.

**Figure 3 antioxidants-11-00683-f003:**
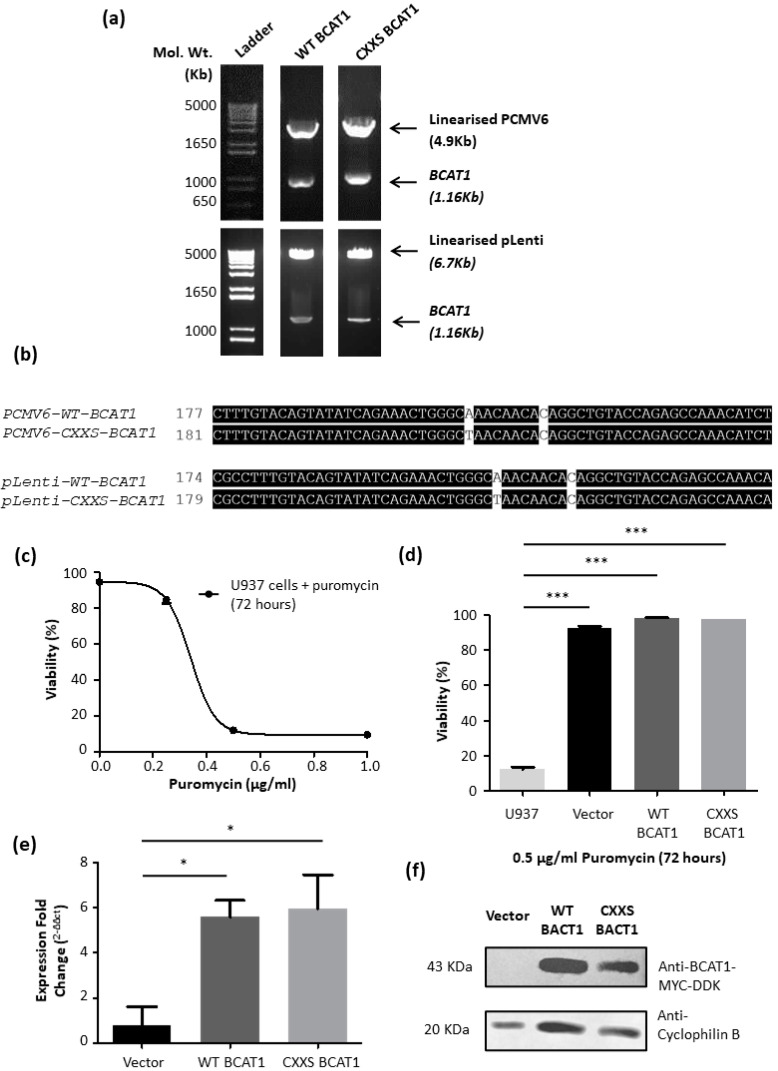
Subcloning *BCAT1* cDNA to pLenti vector and overexpression of BCAT1 in U937 cells. (**a**) Agarose gel image displaying subcloning of *WT BCAT1* and *CXXS BCAT1* cDNA from pCMV6-Entry to pLenti vector. Successful insertion of *BCAT1* was confirmed by the presence of a 1.16 Kb band. (**b**) Sequence alignment of pCMV6-*BCAT1* and pLenti-*BCAT1* using M-Coffee (OmicX, Le Petit-Quevilly, France), CXXC→CXXS base substitutions are highlighted in white, as confirmed by capillary sequencing. (**c**) 72 h dose-response curve to determine the minimum inhibitory concentration of puromycin for U937 cells. (**d**) Graph displaying cell viability and selection of pLenti vector transformed U937 cells following 0.5 µg/mL puromycin treatment and 72 h incubation. (**e**) Graph displaying fold change in *BCAT1* expression in transgenic U937 cells compared to U937 control cells. (**f**) Western Blot analysis showing BCAT1 protein expression in transgenic U397 whole cell lysate compared to Cyclophilin B housekeeping control gene and Vector control cells. Data presented are mean ± SD. Significant differences were calculated using 1-way ANOVA with Tukey’s post-test. * *p* < 0.05, *** *p* < 0.001 (*n* = 4).

**Figure 4 antioxidants-11-00683-f004:**
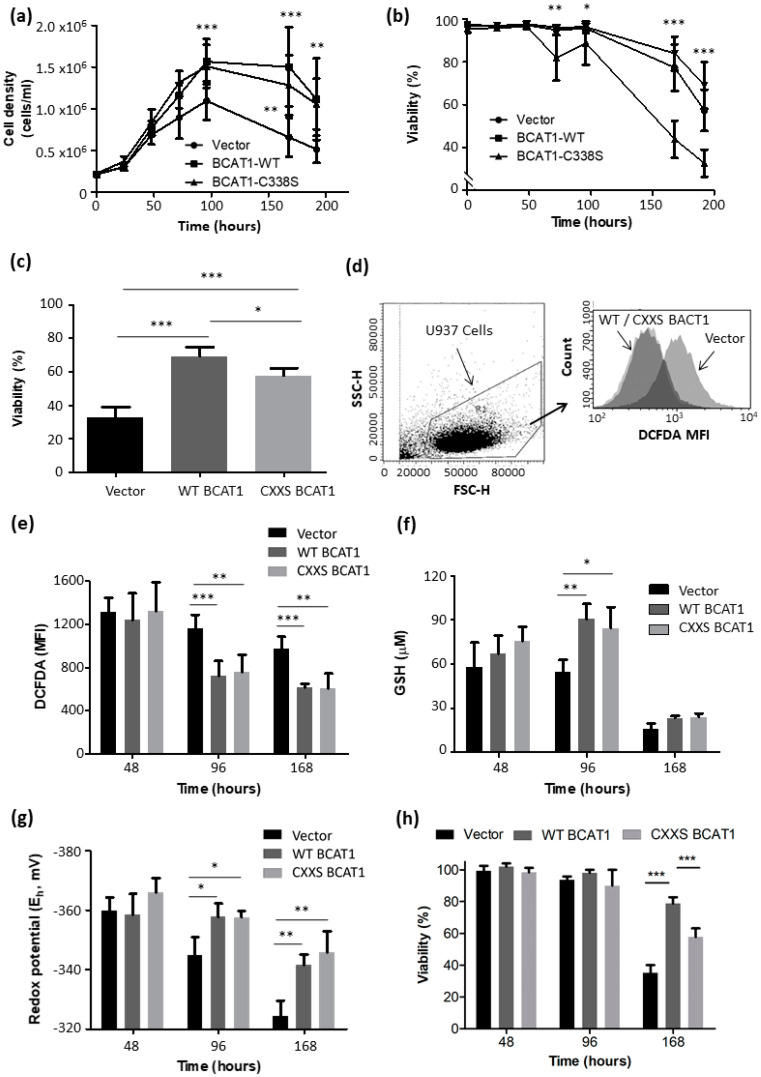
BCAT1 increases growth survival and is associated with altered cellular redox homeostasis in U937 cells. (**a**) Cell density and (**b**) cell viability profile of U937 cells overexpressing WT BCAT1, CXXS BCAT1, and Vector control U937 cells, monitored growth over 0–192 h (*n* = 4). Cells were seeded at 2 × 10^5^ cells/mL at day 0. (**c**) Bar chart displaying cell viability after 192 h. (**d**) Gating strategy DCFDA analysis for intracellular ROS by flow cytometry. (**e**) Intracellular ROS measured by DCFDA for WT BCAT1, CXXS BCAT1 and Vector control U937 cells at exponential (48 h), stationary (96 h), and death phase (168 h). (**f**) Total glutathione (GSH) content for WT BCAT1, CXXS BCAT1 and Vector control U937 cells at exponential (48 h), stationary (96 h), and death phase (168 h). (**g**) Cellular redox potential at exponential (48 h), stationary (96 h), and death phase (168 h). (**h**) cell viability at exponential (48 h), stationary (96 h), and death phase (168 h). Data presented are mean ± SD. Significant differences were calculated using 2-way ANOVA with Bonferroni post-test where, * *p* < 0.05, ** *p* < 0.01 and *** *p* < 0.001 (*n* = 3).

**Figure 5 antioxidants-11-00683-f005:**
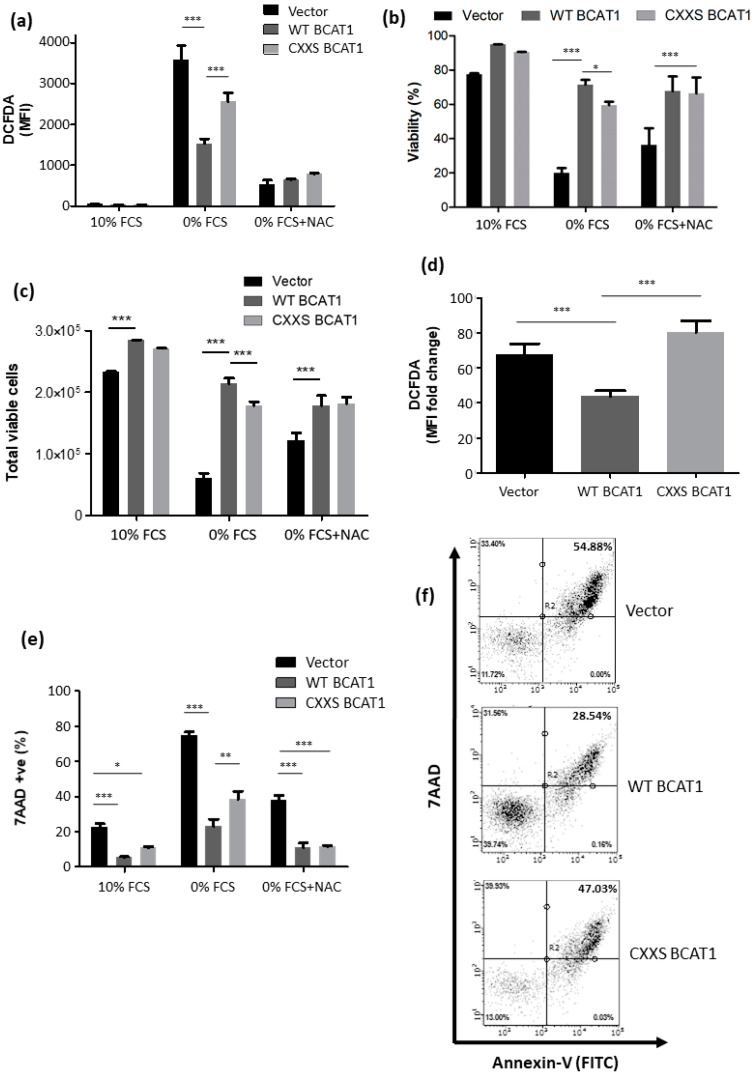
BCAT1 CXXC motif increases survival and modulates intracellular ROS following serum deprivation induced oxidative stress in U937 cells. Serum deprivation-induced by incubation with RPMI ± 10% Foetal Calf Serum (FCS) + 10 mM N-acetyl cysteine (NAC) for 72 h. (**a**) Bar chart displaying DCFDA serum deprivation and NAC treatment at 72 h. (**b**) Cell viability following serum depletion and NAC treatment at 72 h. (**c**) Cell density analysis following serum depletion and NAC treatment at 72 h. (**d**) Fold change in DCFDA signal for Vector Control, WT BCAT1, and CXXS BCAT1 cells following serum deprivation for 72 h. (**e**) Bar chart displaying late apoptotic cells measured by flow cytometry at 72 h. (**f**) Representative flow cytometric plots displaying viable (lower left quadrant), early apoptotic (Annexin-V positive, upper left quadrant), and late apoptotic (7AAD positive, upper right quadrant) cells at 72 h. Data presented are mean ± SD. Significant differences were calculated using 2-way ANOVA with Bonferroni post-test where * *p* < 0.05, ** *p* < 0.01, and *** *p* < 0.001 (*n* = 3). See Appendix A for the NAC dose-response and Appendix A for the trends at 48 h.

**Figure 6 antioxidants-11-00683-f006:**
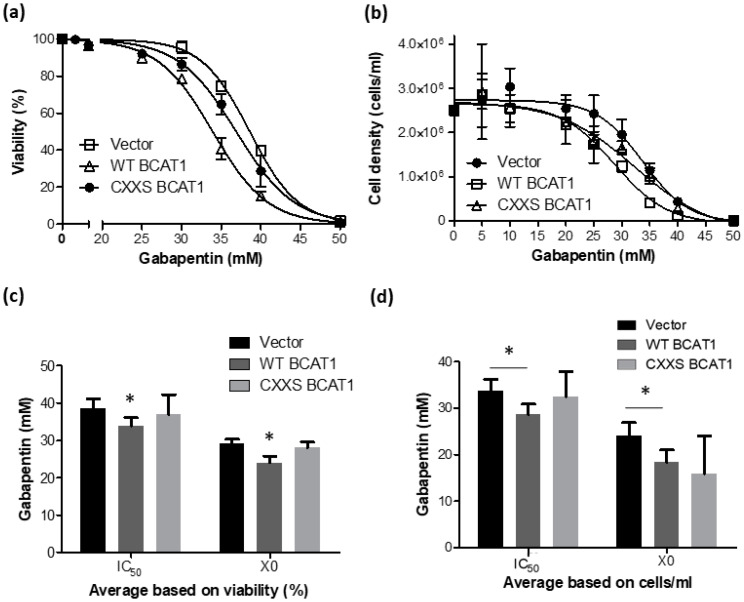
The effect of Gabapentin on Vector Control, WT BACT1, and CXXS BACT1 U937 cells. Vector Control, WT BCAT1, and CXXS BCAT1 overexpressing U937 cells were incubated with Gabapentin at increasing concentrations and monitored for cell density and cell viability. (**a**) Data illustrate the IC_50_ for Gabapentin with respect to cell viability (%) and (**b**) cell density. (**c**) Direct comparison of IC_50_ and minimum inhibitory concentration (X0) for each cell line as determined by viability (%). (**d**) Direct comparison of IC_50_ and X0 for each cell line as determined by cell density. Data presented are mean ± SD (*n* = 3). Significant differences were calculated using 1-way ANOVA with Tukey’s post-test. * *p* < 0.05 (*n* = 3).

**Figure 7 antioxidants-11-00683-f007:**
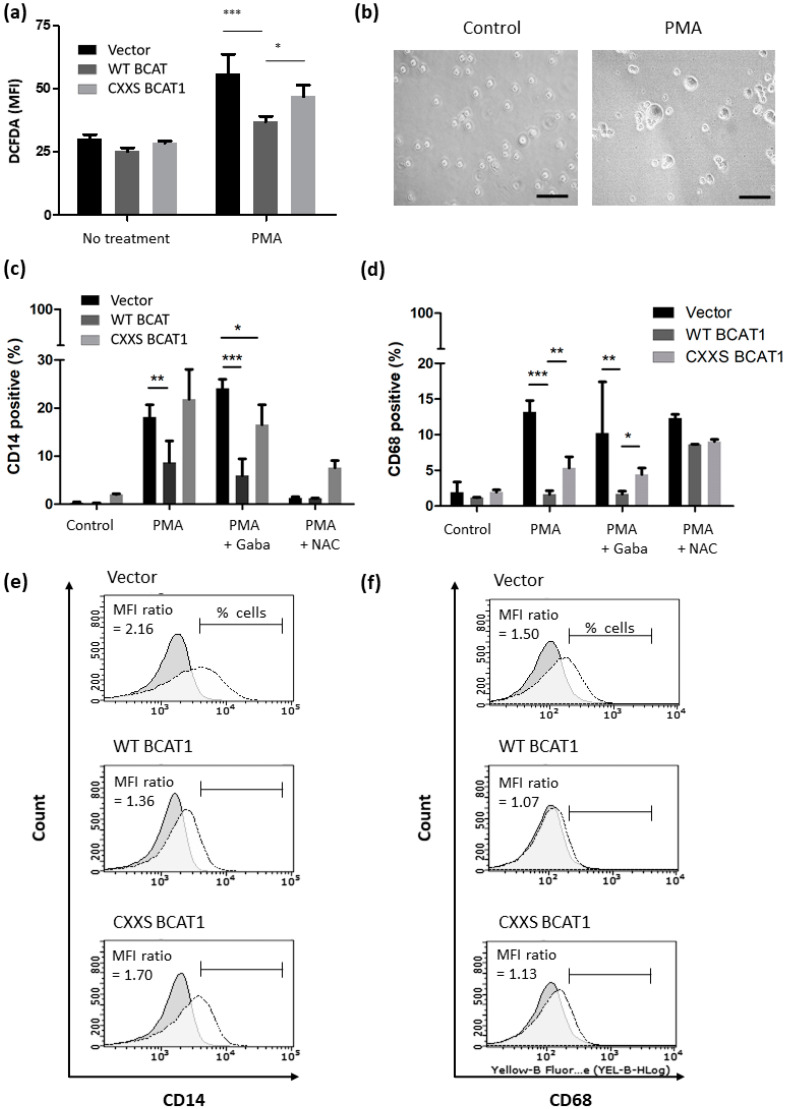
Differential myeloid cell marker expression (CD14 and CD68) and intracellular ROS following treatment with PMA. U937 monocyte to macrophage differentiation was induced by incubation with ±10 ng/mL PMA for 48 h, to control for BCAT1 metabolic and CXXC motif antioxidant, activity cells were supplemented with 20 mM Gabapentin and 10 mM N-acetyl cysteine (NAC) respectively. (**a**) Chart displaying DCFDA MFI ± PMA for WT BCAT1, CXXS BCAT1, and Vector control U937 cells. (**b**) Light microscope Vector U937 cells illustrate a macrophage-like morphology following PMA treatment. Charts displaying percentage; (**c**) CD14 positive cells, (**d**) CD68 positive cells with (**e**) representative flow cytometric histograms for CD14 expression and (**f**) CD68 expression. Data presented are mean ± SD. Significant differences were calculated using 2-way ANOVA with Bonferroni post-test where, * *p* < 0.05, ** *p* < 0.01 and *** *p* < 0.001 (*n* = 3). See Appendix A for flow cytometric gating strategy.

**Figure 8 antioxidants-11-00683-f008:**
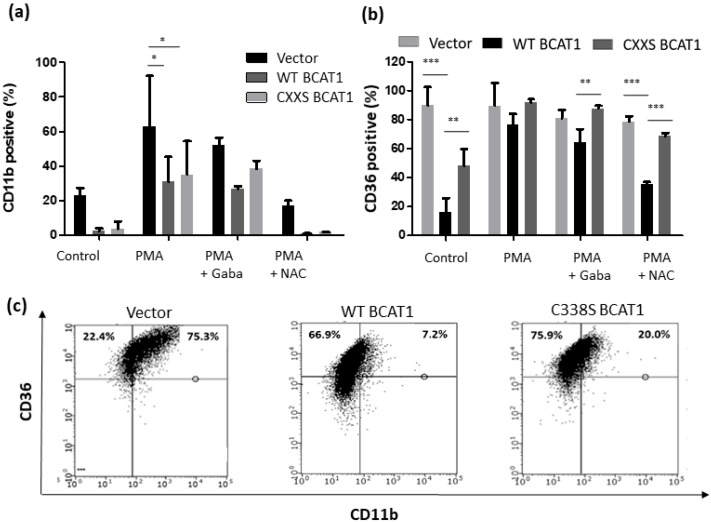
Differential myeloid cell marker expression (CD11b and/CD36) and intracellular ROS following treatment with PMA. U937 monocyte to macrophage differentiation was induced by incubation with ± 10 ng/mL PMA for 48 h, to control for BCAT1 metabolic and CXXC motif antioxidant activity cells were supplemented with 20 mM Gabapentin and 10 mM N-acetyl cysteine (NAC), respectively. (**a**) charts displaying CD11b positive cells and (**b**) % CD36 positive cells as measured by flow cytometry. (**c**) Representative flow cytometric bivariate plots displaying % CD36 positive cells (y-axis) and % CD11b positive cells (x-axis) across all cell lines following treatment with PMA. Data presented are mean ± SD. Significant differences were calculated using 2-way ANOVA with Bonferroni post-test where, * *p* < 0.05, ** *p* < 0.01 and *** *p* < 0.001 (*n* = 3). See Appendix A for flow cytometric gating strategy.

**Figure 9 antioxidants-11-00683-f009:**
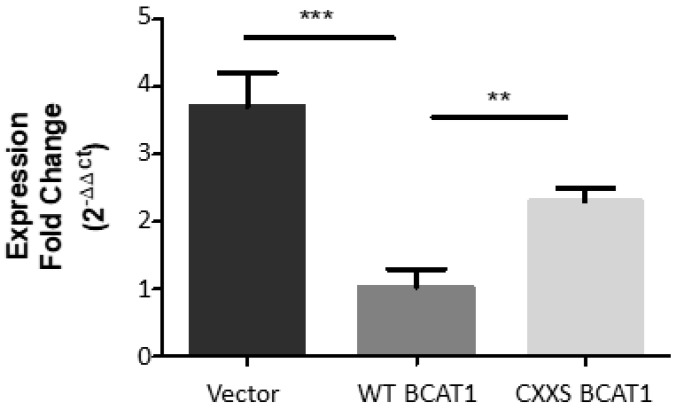
Thioredoxin Reductase (TrxRD1) relative gene expression in U937 cells. Graph displaying fold change in *TrxRD1* expression in transgenic U937 cells. Significant differences were calculated using 1-way ANOVA with Tukey’s post-test where, ** *p* < 0.01 and *** *p* < 0.001 (*n* = 3).

**Table 1 antioxidants-11-00683-t001:** Gabapentin IC_50_ (mM) and X0 (mM) for Transformed U937 Cells. IC_50_ was determined by non-linear regression using Log10 [Gabapentin]. Minimum inhibitory concentration (X0) was determined by non-linear regression, using the plateau followed by one phase death decay model. Respective R2 values are presented below.

	Vector	WT BCAT1	CXXS BACT1
**IC_50_ based on viability (%)**	38.6 ± 0.30 mM	33.92 ± 0.22 mM	36.88 ± 0.36 mM
**R^2^**	0.98	0.991	0.975
**IC_50_ based on density (cell/mL)**	33.77 ± 1.19 mM	28.67 ± 1.07 mM	32.41 ± 2.61 mM
**R^2^**	0.919	0.947	0.875
**X0 based on viability (%)**	29.01 ± 0.60 mM	24.38 ± 0.72 mM	28.65 ± 0.79 mM
**R^2^**	0.983	0.97	0.966
**X0 based on density (cell/mL)**	23.99 ± 1.38 mM	18.35 ± 1.29 mM	15.89 ± 3.91 mM
**R^2^**	0.908	0.934	0.867

## Data Availability

All of the data is contained within the article and the Appendix A.

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
