# Peer review of "The BCAT1 CXXC Motif Provides Protection against ROS in Acute Myeloid Leukaemia Cells"

_antioxidants, 2022, doi:10.3390/antiox11040683_

Round 1

Reviewer 1 Report

The Authors showed the role of the BCAT1 CXXC motif in AML and to understand how this may influence cellular redox processes.

The manuscript is well written, the results support the hypotheses. The figures should be improved in quality so the work would take on another aspect. Figure 7b is not very credible, when U937 become macrogphages adhere to the plate and assume the morphology of a macrophage. the figure is not clear and not well visible. The authors should also show the flow cytometers panels related to cd68 and cd14. This figure does not convince me very much.

Author Response

Thank you again for taking time to review our manuscript and provide valuable constructive feedback. We agree with your comments and have addressed these to the best of our ability in the revised manuscript. In order to address your feedback fully, we felt it necessary to split Figure 7.
This has enabled us to accommodate the additional representative flow cytometric histograms displaying CD14 and CD68 expression changes. The data contained in the original Figure 7 is now split between Figure 7 (deals with CD14 and CD68 only) and Figure 8 (deals with CD11b and CD36).
Changes highlighted in yellow on the resubmission.

In response to your comments:

Fig 7b. We have removed the Giemsa images and replaced with phase contrast images to better show cell morphology under our light microscope. We also include representative flow cytometric overlayed histograms, which display CD14 and CD68 expression changes between our Vector control and WT / CXXS BCAT1 cells. Included on these histograms is the mean fold MFI change (ratio).

Reviewer 2 Report

Authors addressed all my requests and suggestions to my fullest satisfaction.
Small comment, the supplementary Fig. S4B would need some compensation work....this would be better to correct before final approval for publication ( up to editor).
Also the gating strategy on Fig. 4d ( is going in right direction), what I am missing here is the exclusion of dead cells?).
Overall, congratulations on this work.

Author Response

Thank you again for taking time to review our manuscript and provide valuable constructive feedback, which has improved the quality of our work submitted.

In response to your questions:

Fig S4B. This is fully compensated. The black population illustrates the isotype control for the FITC and PE conjugated antibodies. These isotype controls were used to set the negative thresholds, as indicated by the quadrants on the plot. The red population shows positive binding of anti-CD36 and anti-CD11b to the U937 cells (gated on the previous plot S4A, FSC/SSC).
The curve in the plot is not due to a compensation issue, it is demonstrating the up-regulation of CD36, which is followed by up-regulation of CD11b.

Fig 4d. This is going in the correct direction. The DCFDA histogram data presented to the right is based on the viable homogeneous U937 cell gated population, which was determined by forward scatter (FSC) and side scatter profile (SSC) in the left hand plot. Debris in the bottom left of this FSC/SSC plot includes dead cells, which are eliminated and not considered for DCFDA analysis.

Reviewer 3 Report

The manuscript is clear and well written. The data presented by the authors are robust and interesting but are based on a single AML cell line, which is not a bonafide AML cell line (derived from the pleural effusion of a 37-year-old, White, male patient with histiocytic lymphoma, according to ATCC). Thus, authors should discuss limitations in interpreting their findings.

Author Response

Thank you for taking time to review are work and for giving us the opportunity to comment on the use of U937 cells as a limitation for the work presented. We have included a section dedicated to addressing this in the discussion section, along with supporting references. This additional narrative is added at line 645-660 and highlighted in yellow on the resubmission.

We have included what we have written below for your convenience:

“The U937 cell line was originally isolated from the pleural effusion of a patient with histiocytic lymphoma [42]. Early characterisation studies showed that U937 cells differed from typical lymphoblastoid cell lines [42]. Because of their pro-monocytic phenotype, many regard the U937 cell line to be representative of an AML subtype M5, under the French-American-British (FAB) classification system [43–46]. This pro-monocytic phenotype and cytochemistry, has accounted for the wide use of U937 cells as a model to study AML, including ROS mediated monocyte to macrophage transition [32,49,50]. We identified U937 cells as having a relatively low basal expression of BCAT1 compared other myeloid cell lines [49], which would allow an effective comparison between BCAT1 overexpressing cells to the vector control. Taken together, the U937 cell line presented as a good cellular model to study the putative antioxidant effect of the BCAT1 CXXC motif in myeloid differentiation, which is a defining feature of AML [50]. However, our study is limited to a single FAB subtype model. Future work could extend our analysis and examine the antioxidant capacity of the BCAT1 CXXC motif in other FAB subtype models, for example HL60 (AML, M2) and NB4 (AML M3 aka APL) [45].”

Round 2

Reviewer 1 Report

The authors have now addressed all my major initial comments.